# Knowledge, Attitude, and Practice of Indonesian Residents toward COVID-19: A Cross-Sectional Survey

**DOI:** 10.3390/ijerph18094473

**Published:** 2021-04-23

**Authors:** Muhammad Muslih, Henny Dwi Susanti, Yohanes Andy Rias, Min-Huey Chung

**Affiliations:** 1School of Nursing, College of Nursing, Taipei Medical University, Taipei 11031, Taiwan; muslih@umm.ac.id (M.M.); hanisusanti@yahoo.com (H.D.S.); 2School of Nursing, Faculty of Health Science, University of Muhammadiyah Malang, Malang 65145, Indonesia; 3Faculty of Health and Medicine, College of Nursing, Institut Ilmu Kesehatan Bhakti Wiyata Kediri, Kediri 64114, Indonesia; yohanes.andi@iik.ac.id; 4Department of Nursing, Shuang Ho Hospital, Taipei Medical University, New Taipei City 23561, Taiwan

**Keywords:** knowledge, attitude, practice, COVID-19

## Abstract

Coronavirus disease 2019 (COVID-19) has become a pandemic. We examined the KAP’s relationship with factors associated with practice toward the COVID-19 pandemic in Indonesia. This cross-sectional survey study was conducted between March and April 2020 and included 1033 participants. Knowledge scores of COVID-19 were positively associated with wearing a mask when leaving home (odds ratio (OR): 1.22, *p* < 0.05). Although men had a lower knowledge score, they were less likely to go to a crowded place compared with women (OR: 0.79, *p* < 0.05). However, women (OR: 1.25, *p* < 0.05) were more likely than men to wear a mask when leaving home. Furthermore, men (OR: 3.32, *p* < 0.05) were more likely than women to have a positive attitude toward COVID-19. Indonesian residents had satisfactory knowledge, demonstrated a positive attitude, and followed appropriate practices toward the pandemic. More educated individuals had a more positive attitude. Men and women differed with respect to their knowledge-based practices. Men were less likely to go to crowded places, and women were more likely to wear a mask when leaving home. Furthermore, men were more likely to wear a mask when leaving home than women when men had the attitude that Indonesia can win against COVID-19.

## 1. Introduction

Coronavirus disease 2019 (COVID-19) has become a pandemic; the virus spreads rapidly and is associated with high morbidity and mortality rates [1]. According to the World Health Organization (WHO), as of 24 February 2021, a total of 111,762,965 cases and 2,479,678 deaths of COVID-19 were reported in 222 countries and territories, with 12% of cases worldwide (13,415,064) reported in Southeast Asia [2]. This number has considerably increased since the first case was reported on 20 January 2020 [2]. The WHO declared COVID-19 as a global public health emergency in January 2020 and then a pandemic in March 2020 [2]. In Indonesia, COVID-19 was first detected on 2 March 2020, in two women aged 64 and 31 years, and the number of COVID-19 cases continues to increase. On 24 February 2021, the number of COVID-19 cases in Indonesia reached 1,306,141 with 35,254 deaths [3], and Indonesia became the country with the highest number of positive cases in Southeast Asia [2]. The spread of COVID-19 in the population to date can be evaluated using an epidemiological model. For example, a previous study estimated the parameters of an epidemiological model to model the numbers of suspected, infected, and recovered cases of COVID-19 and COVID-19-related deaths in Italy by using a dynamic graph based on machine learning [4].

Evidence indicates that the human-to-human transmission of COVID-19 occurs through respiratory droplets (e.g., sneezing or coughing) and through contact with the secretions of infected people [1,5,6,7,8]. On 9 July 2020, the WHO reported that the possible modes of the transmission of severe acute respiratory syndrome coronavirus 2, the causative agent of COVID-19, include contact, droplet, airborne, fomite, fecal–oral, bloodborne, mother-to-child, and animal-to-human transmission [9]. Given that the transmission mechanism of COVID-19 is similar to that of common flu or influenza virus, including community transmission, preventive measures should be immediately implemented to minimize transmission risk. Community transmission has been the main cause of the high number of COVID-19 cases in Indonesia [2]. Therefore, implementing measures to control COVID-19′s spread and timely assessments of public awareness are urgently required, particularly considering that compared with other countries, Indonesia has no previous experience of managing a pandemic.

Personal hygiene measures, such as washing hands, wearing a mask, and following respiratory etiquette, can help reduce infection transmission [10,11,12]. Reducing contact, increasing social distancing, and improving personal hygiene can mitigate the community spread of COVID-19 [13]. These actions should be promoted and considered to be main precautions against COVID-19 to prevent its spread. Indonesia’s Ministry of Health has been providing updates and health protocols toward COVID-19 through its website, social media, and live reports since the first set of positive cases was detected in March 2020 [14]. In addition, the Ministry of Health has implemented policies to prevent and mitigate the spread of COVID-19 in public areas and health protocols for Indonesians or foreign nationals arriving from overseas into quarantine area where social distancing has been mandated [15].

To successfully control and reduce the spread of COVID-19, changes are required in those behavioral practices that are influenced by public knowledge and attitudes. Assessing public knowledge is critical to identifying gaps and strengthening ongoing prevention efforts [16]. Knowledge is essential for changing one’s attitudes and practices [17], particularly during a pandemic. A study revealed that the lack of knowledge contributes to unfavorable attitudes and practices, which exert negative effects on infection control [18]. This might be because knowledge is the main determinant of positive attitudes toward COVID-19 prevention, which can be implemented after knowing what activities ought to be performed [19]. Thus, by gaining knowledge, awareness regarding public attitudes and practices toward COVID-19 can be achieved. In other words, knowledge can be used to identify factors that can help the general public have a positive attitude and implement healthy practices toward COVID-19 prevention.

The success of the fight against COVID-19 depends on public attitudes [16]. Some studies have reported positive and optimistic attitudes among the general public toward overcoming COVID-19 in countries such as China [20], Malaysia [21], Saudi Arabia [16], and Ethiopia [19]; specifically, residents in these countries are confident that COVID-19 can be handled appropriately. Positive attitude has been correlated with adequate knowledge [22,23]. Therefore, gaining adequate knowledge can lead to positive attitudes and better responses toward COVID-19.

Although information on COVID-19 is abundant, the knowledge held does not correspond with desirable attitudes and practices [23]. Many people do not comply with the government’s recommendations of staying home, social distancing, wearing masks, and washing their hands. The experience of other countries during a pandemic [24,25] indicates that general public awareness is essential for guaranteeing the control of the COVID-19 outbreak, and this awareness is affected by the knowledge, attitude, and practice (KAP) of a population [20]. KAP toward COVID-19 determines a society’s readiness to accept the behavioral changes entailed by measures promulgated by health authorities [21]. Therefore, practice toward COVID-19 can be affected by knowledge and attitudes.

A study reported that both knowledge and attitude are significant predictors of practice or behavior toward COVID-19 [26]. Some studies conducted in Egypt [27] and Malaysia [21] have examined the prevalence of COVID-19 and the relationship between demographic factors and KAP; however, they did not assess the crucial factor of predicted practice. By contrast, a study conducted in China [20] evaluated factors that predict practice but did not fully examine knowledge and attitude. A study on knowledge and attitudes toward COVID-19 was conducted in Indonesia [23]. The authors found an albeit weak correlation between knowledge and attitudes toward COVID-19. Furthermore, this previous study did not measure the concept of practice, and its sample size was too small to represent the general public affected by COVID-19 in Indonesia. Therefore, the relationship among knowledge, attitudes, and practices toward COVID-19 to date should be investigated in Indonesia, considering that Indonesia is one of the countries in Asia with the highest number of COVID-19 cases.

A previous study referred to the KAP model, which is based on the premise that having more knowledge changes one’s behavior [28]. Therefore, in this study, we examined the relationship between KAP and factors that were identified to predict practices toward the COVID-19 pandemic among Indonesian residents, which comprised a large sample size that was representative of the Indonesian population.

## 2. Methods

### 2.1. Study Design

This cross-sectional survey conducted between 25 March 2020–30 April 2020, examined the demographic characteristics and KAP of Indonesian residents toward COVID-19. This study collected data through an Internet questionnaire. The survey was conducted using a Google Form link shared on social networking sites, such as WhatsApp and Facebook, which are the most accessible social media platforms in Indonesia. Initially, the survey was distributed through a WhatsApp group, and participants were invited to complete the form. A previous study proposed a novel summarization technique for creating multimedia stories by applying social media content to online social networks [29].

Internet questionnaires constitute a methodological alternative in epidemiological data collection [30]. We used this method because it was impossible to directly retrieve data during the COVID-19 pandemic. In the first section of the survey, we described the purpose of this study and provided details regarding informed consent. If participants were willing to participate in this study, they were asked to fill the consent form and then directed to the online questionnaire. This study was approved by the Health Research Ethics Committee of Institute of Health Science STRADA Indonesia (No: 1911/KEPK/IV/2020).

### 2.2. Participants

Individuals who were aged ≥17 years at the time of the study, could read Indonesian, and provided informed consent were included. Individuals who had already received a diagnosis of COVID-19 at the time of the study were excluded. The factors that influence the sample size estimation of the study are significance level (α), effect size (ES), and power [31]. In epidemiological studies, ES could be calculated based on odds ratio (OR) where OR of 1.68, 3.47, and 6.71 were equivalent to small, medium and large ES [32]. A previous study also reported where OR less than 1.50 was equal to small ES [32].

In our study, the sample size was calculated using G Power, version 3.1, where the significant level (α) set at, 0.05, OR of 1.37 was obtained from a previous study [20], and power of 0.95. This yielded a total sample size 603. We expected a potential missing data of 20% and thus aimed to recruit at least 723 participants. The sample size was calculated based on estimates from the distribution of the general population as reported by the Central Bureau of Statistics [33]. A proportion from eastern, central, and western regions of Indonesia are reported at 2.76%, 16.14% and 81.10%, respectively [33]. In our study, we reached participants from all regions of Indonesia and obtained 1.4%, 6% and 92.6% from each base, which has a similar pattern to the proportional distribution of these regions in the general population.

Throughout data collection, we excluded 34 participants with duplicate responses. Consequently, we used participants email address to avoid selection bias and overlapping response during data collection. This method was used to maintain security and validity. Finally, 1033 participants constituted the sample in the data analysis. Considering the significance level (α), OR, power, and the distribution of the sample size by region, our sample size was adequate.

### 2.3. Instrument

The questionnaire comprised two parts: demographics and KAP. The demographic variables included age, sex, marital status, education, occupation, and region. The KAP questionnaire used in this study was developed by [20]. The original KAP scale was developed in China and is available in English. The knowledge scale included 12 questions: four on clinical signs (K1–K4), three on transmission routes (K5–K7), and five on prevention and control (K8–K12). The three possible answers were “true,” “false,” and “I don’t know.” A correct answer was assigned a score of 1, and the other two options were assigned a score of 0. Thus, the total knowledge score ranged from 0 to 12, with higher scores indicating better knowledge of COVID-19. The Cronbach’s alpha of the original questionnaire was 0.71 [20], indicating acceptable internal consistency and reliability [34]. Attitude toward COVID-19 was measured using two questions: will the pandemic be controlled successfully? (possible answers were “agree,” “disagree,” or “I don’t know”); and: are you confident that Indonesia can win the battle against COVID-19? (possible answers were “yes” or “no”). Practice was examined by asking participants whether they go to crowded places and wear a mask when leaving home, and the possible answers were “yes” or “no” (yes = 1; no = 0).

After obtaining permission from the original authors, the original questionnaire was independently translated into Indonesian by using the forward and back translation approach. Two translators, a certified translator and an expert in nursing and research in Indonesian universities, whose first language was Indonesian and who were bilingual and fluent in English translated the questionnaire. The translators assessed whether the questionnaire items were relevant for precisely measuring KAP toward COVID-19 and ensured linguistic and conceptual equivalence. Initially, a pretest was conducted before the distribution of the Indonesian version of the KAP questionnaire to the study sample.

Exploratory factor analysis (EFA) was performed to examine construct validity. The Kaiser–Meyer–Olkin (KMO) and Bartlett’s tests of sphericity were used to determine and verify sampling adequacy [35,36]. The KMO index ranges from 0 to 1 [37], and the recommended value is >0.60 [38]. Bartlett’s test of sphericity is passed when significance at *p* < 0.001 is achieved [38]. A factor loading of >0.40 was used to identify whether the item scale could represent its factor [39]. Item discriminant analysis reflects how much an item represents a particular construct. The results of this analysis are considered to be satisfactory if the index is >0.30 [40]. In our study, the KMO value was 0.68 and Bartlett’s test of sphericity value was significant (*p* < 0.001), indicating sampling adequacy for factor analysis. The factor loading of all items was >0.40. Furthermore, the findings of item discriminant analysis were significant (*p* < 0.001), with the index values of the most items being >0.30, indicating the reliability of the scale.

### 2.4. Statistical Analysis

Statistical analyses were performed using SPSS (version 23; IBM, Armonk, NY, USA). Descriptive statistics were used to present demographic characteristics. Continuous variables are presented as their mean ± standard deviation and categorical variables are presented as the frequency (*n*) and percentage (%). Bivariate analyses such as the independent *t* test, chi-square test, and analysis of variance were performed to examine the relationship between demographic characteristics and KAP. Hierarchical logistic regression was conducted to examine factors associated with practice toward COVID-19. Multivariate logistic regression was used to investigate the predictive effect of practice toward COVID-19. Statistical significance was indicated by *p* < 0.05.

## 3. Results

Table 1 lists the descriptive analysis of KAP scale. The mean and SD of knowledge score was 10.00 (1.44). The demographic characteristics of study participants and knowledge score of COVID-19 are shown in Table 2. Most participants were aged between 17 and 29 years (53.40%) and were predominantly women (694, 67.20%), and more than half of the participants were unmarried (521, 50.40%). One-third of study participants had a bachelor’s degree (390, 37.80%), and almost half of them were employed (482, 46.70%). Most participants were living in the western region of Indonesia (957, 92.60%). Significant differences in knowledge scores and demographic characteristics were observed among different age groups (t = 6.07, *p* < 0.001), marital statuses (t = −5.46, *p* < 0.001), educational levels (F = 13.61, *p* < 0.001), and occupations (F = 14.89, *p* < 0.001). A lower educational level (middle school and below; β = −0.13, *p* < 0.05) was significantly associated with a lower knowledge score.

Table 3 shows the association between demographic characteristics and attitude. Sex was significantly associated with the attitude that COVID-19 will be successfully controlled (*p* < 0.05) and Indonesia can win the battle against COVID-19 (*p* < 0.05). Knowledge score was significantly associated with the attitude that COVID-19 will be successfully controlled (*p* < 0.001). Table 4 presents the association between demographic characteristics and practice. Sex (*p* < 0.05), educational level (*p* < 0.05), and occupation (*p* < 0.05) were significantly associated with the practice of going to a crowded place, and sex (*p* < 0.001), educational level (*p* < 0.05), and knowledge scores (*p* < 0.05) were significantly associated with the practice of wearing a mask when leaving home.

Table 5 shows the results of a hierarchical logistic regression analysis of factors significantly associated with practice. In both step 1 (OR: 0.71, *p* < 0.05) and step 2 (OR: 0.72, *p* < 0.05), a positive and significant association was observed between women and the practice of going to a crowded place during a pandemic. However, this association was not found to be significant in step 3. Women were more likely to demonstrate the attitude of wearing a mask when leaving home compared with men (step 1; OR: 2.11, *p* < 0.001; step 2; OR: 2.03, *p* < 0.05; step 3; OR: 1.92, *p* < 0.05). Female sex was the strongest significant predictor of the aforementioned attitude. Nevertheless, the knowledge score (OR: 0.90, *p* < 0.05) was significant in step 2. Knowledge score was positively associated with the practice of wearing a mask when leaving home in steps 2 (OR: 1.26, *p* < 0.001) and 3 (OR: 1.22, *p* < 0.05). Knowledge regarding COVID-19 was found to be a significant predictor of the attitude of wearing a mask when leaving home.

Table 6 shows the results of the multivariate logistic regression performed to predict practice toward COVID-19 among men and women. With respect to knowledge score, men were less likely to practice going to a crowded place compared with women (OR: 0.79, *p < 0.05*). The finding indicated that male sex was a significant predictor of the practice of going to a crowded place according to the knowledge score of COVID-19. Moreover, women (OR: 1.25, *p* < 0.05) were more likely to practice wearing a mask when leaving home compared with men. Hence, female sex was a significant predictor of this practice. With respect to the attitude that Indonesia can win the battle against COVID-19, men (OR: 3.32, *p* < 0.05) were more likely to practice wearing a mask when leaving home compared with women.

## 4. Discussion

To the best of our knowledge, no study has analyzed a population’s KAP toward COVID-19, particularly in Indonesia. COVID-19 is a relatively new disease, and Indonesia has no prior experience in managing an outbreak of this scale. Although a previous study examined knowledge and attitude, it did not evaluate practices toward COVID-19 [23]. Our study contributes to the literature by including the variable of practice. The strength of the present study is its large sample size comprising participants recruited from six residential areas in Indonesia, making it representative of the Indonesian population. Our results indicated sex differences in the relationship between demographic characteristics and KAP toward COVID-19 among Indonesian residents. Furthermore, we examined factors that predicted practice toward the COVID-19 pandemic in Indonesia.

Participants aged ≥30 years had a higher knowledge score (Table 1). This result is similar to that reported in studies conducted in China [20], Malaysia [21], and India [41]. Our study results revealed that married people had better knowledge, which coincides with the finding of a study conducted in China [42]. Married people had more knowledge regarding the COVID-19 pandemic possibly because they are responsible to care for both themselves and their family members. Furthermore, people with a higher education level and those who were employed as a teacher had higher knowledge scores. This finding is consistent with the assumption that most participants who had a higher educational level have better knowledge regarding COVID-19 among public and health care professionals; this finding is also supported by those of studies conducted in China [20,42], India [41], and other countries [43,44,45]. In addition, most participants in our study had adequate knowledge regarding COVID-19. Since the first Indonesian case of COVID-19 was detected in March 2020, information regarding this disease has been widely reported worldwide; therefore, Indonesians had satisfactory knowledge regarding COVID-19. Our finding is in line with that of a previous study conducted in Indonesia that reported that most of the participants had satisfactory knowledge regarding COVID-19 [23]. Women were found to have higher knowledge scores than did men; this finding is similar to those of studies conducted in China [46] and Bangladesh [47], although the result was nonsignificant. This finding may be attributed to the fact that female participants in our study had a higher educational level. In addition, a previous study reported that women had a higher literacy level compared than did men in terms of the prevention and control of infectious diseases [47].

Most of our participants exhibited a positive attitude toward COVID-19. More than 90% of participants agreed that COVID-19 will be successfully controlled and believed that Indonesia can win against COVID-19. These results are similar to those of previous studies conducted in Indonesia [23] and elsewhere [26,42,48,49,50]. These results indicated that although COVID-19 has spread worldwide, most people had an optimistic attitude; they believed that the COVID-19 pandemic will end and that their countries will win against this pandemic. Among demographic characteristics, sex was significantly associated with the attitude that COVID-19 will be successfully controlled, and Indonesia can win the battle against COVID-19 (Table 2). Similarly, a previous study reported that sex was significantly associated with attitude [49]. Moreover, higher knowledge scores were significantly associated with the positive attitude that COVID-19 will be successfully controlled. This finding is consistent with those of previous studies conducted in China [20,46], Malaysia [21], and other studies [48,50]. These findings imply that improving knowledge regarding COVID-19 can affect the attitude toward it and vice versa. In our recent study, the confidence of participants that COVID-19 will be successfully controlled may be because of actions undertaken by the Indonesian government against COVID-19.

The results of this study revealed that most participants followed appropriate practices such as not going to crowded places and wearing a mask when leaving home (Table 3). Thus, our study participants appeared to be willing to follow good practices against the COVID-19 pandemic. Occupation and knowledge scores were found to be significantly and positively associated with the practice of going to a crowded place and wearing a mask when leaving home, respectively, whereas sex and educational level were significantly and positively associated with both practices. Our findings are inconsistent with those of [51] who found that knowledge is a necessary but insufficient condition in changing individual or collective behavior, thus indicating that the impetus to change practices does not come from knowledge alone. For example, some of the participants in the present study were aware of threats associated with the COVID-19 pandemic; however, they continued with their pre-pandemic practices such as going to crowded places. The present study elucidates the relationship between knowledge and practice among Indonesian residents during the COVID-19 pandemic and the potential effect of attitude on the relationship between knowledge and practice. The findings of this study indicate that encouraging good practices involves building knowledge and fostering a positive attitude toward COVID-19. Moreover, bad practices may occur due to the lack of knowledge and negative attitude. Thus, both knowledge and attitude affect practice regarding COVID-19; this finding is in agreement with KAP theory [52,53], which indicates that changes in practice can result from changes in knowledge and attitude.

In our study, women more likely to wear a mask when leaving home compared with men (Table 4). However, men were also less likely to go to a crowded place compared with women; this result contradicted the finding of a previous study that men were more likely to engage in risk-taking behavior [54]. Our finding is consistent with that of a study conducted in China [20] that found a significant association between men and potentially dangerous practices toward COVID-19, including going to a crowded place during the pandemic or not wearing a mask when leaving home. The pandemic has resulted in vacant public spaces (e.g., schools and offices) because people are conducting their lives remotely (including working or studying) as part of government policy. Regarding KAP toward COVID-19 in Indonesia, our findings indicated that specific interventions that may be required to change KAP. For example, considering that KAP significantly differed between men and women, tailored interventions that target a specific gender are likely to be more effective at behavioral change.

The limitation of this study is that data were collected through social media; bias was possible due to some target populations not being represented. However, similar surveys have been conducted [20,21,27,55] because direct sampling through a community survey is not possible due to social distancing. Another limitation is overlapping response via Google online forms. Google forms was a user-friendly platform survey, but it has limitation of inability to track IP address [56]. A previous study used internet protocol (IP) address to identify potential duplicate accesses from the same user [57]. However, IP address can be shared by a number of people in the same area [58]. Therefore, we did not limit the responses to a certain IP address. Furthermore, a previous study reported access issue was a weakness of online survey research [59]. In addition, considering that the community transmission of COVID-19 remains a problem in Indonesia, additional studies may be required.

## 5. Conclusions

The findings of this study suggested that Indonesian residents had satisfactory knowledge, demonstrated a positive attitude, and followed appropriate practices toward the COVID-19 pandemic. Individuals with a higher educational level demonstrated a positive attitude. Differences were observed in practices followed by men and women with respect to their knowledge. Men were less likely to go to a crowded place, and women were more likely to wear a mask when leaving home. In addition, with respect to the attitude that Indonesia can win against COVID-19, men were more likely to wear a mask when leaving home compared with women.

In summary, our findings demonstrate that the assessment of KAP is vital for evaluating public adherence toward COVID-19. These results can be used to inform the public on how to behave appropriately during the pandemic. The findings of this study may also be helpful for health professionals and policymakers in Indonesia to develop targeted interventions and effective practices and to modify their communication campaigns against COVID-19.

## Figures and Tables

**Table 1 ijerph-18-04473-t001:** Descriptive analysis of knowledge, attitude, and practice variables (*n* = 1033).

Variable				
Knowledge		Correct ^a^	Incorrect ^a^	
K1. The main clinical symptoms of COVID-19 are fever, fatigue, dry cough, and myalgia.		915 (88.60)	118 (11.40)	
K2. Unlike the common cold, stuffy nose, runny nose, and sneezing are less common in persons infected with the COVID-19 virus.		694 (67.20)	339 (32.80)	
K3. There currently is no effective cure for COVID-2019, but early symptomatic and supportive treatment can help most patients recover from the infection.		889 (86.10)	144 (13.90)	
K4. Not all persons with COVID-2019 will develop to severe cases. Only those who are elderly, have chronic illnesses, and are obese are more likely to be severe cases.		831 (80.40)	202 (19.60)	
K5. Eating or contacting wild animals would result in the infection by the COVID-19 virus.		421 (40.80)	612 (59.20)	
K6. Persons with COVID-2019 cannot infect the virus to others when a fever is not present.		855 (82.80)	178 (17.20)	
K7. The COVID-19 virus spreads via respiratory droplets of infected individuals.		970 (93.90)	63 (6.10)	
K8. Ordinary residents can wear general medical masks to prevent the infection by the COVID-19 virus.		751 (72.70)	282 (27.30)	
K9. It is not necessary for children and young adults to take measures to prevent the infection by the COVID-19 virus.		960 (92.90)	73 (7.10)	
K10. To prevent the infection by COVID-19, individuals should avoid going to crowded places such as train stations and avoid taking public transportations.		1007 (97.50)	26 (2.50)	
K11. Isolation and treatment of people who are infected with the COVID-19 virus are effective ways to reduce the spread of the virus.		1018 (98.50)	15 (1.50)	
K12. People who have contact with someone infected with the COVID-19 virus should be immediately isolated in a proper place. In general, the observation period is 14 days.		1017 (98.50)	16 (1.50)	
**Total score of Knowledge**	10.00 (1.44) ^b^			
**Attitude**				
A1. Do you agree that COVID-19 will finally be successfully controlled?		Agree ^a^ 974 (94.28)	Disagree ^a^ 14 (1.36)	Don’t know ^a^ 45 (4.36)
A2. Do you have confidence that Indonesia can win the battle against COVID-19 virus?		Yes ^a^ 986 (95.5)	No ^a^ 47 (4.50)	
**Practice**				
P1. In recent days, have you gone to any crowded place?		Yes ^a^ 243 (23.50)	No ^a^ 790 (76.50)	
P2. In recent days, have you worn a mask when leaving home?		Yes ^a^ 912 (88.30)	No ^a^ 121 (11.17)	

^a^ n (%); ^b^ mean (standard deviation).

**Table 2 ijerph-18-04473-t002:** Demographic characteristics and knowledge scores of COVID-19 (*n* = 1033).

Characteristics	*n* (%)	Knowledge Score (Mean ± SD)	t ^a^/F ^b^	*p* Value	β-Coef (95% CI)	*p* Value
Age (years)						
17–29	552 (53.4)	9.75 ± 1.58	6.07	<0.001	−0.08 (−0.47–0.04)	0.090
≥30	481 (46.6)	10.28 ± 1.19			Ref	
Sex						
Women	694 (67.2)	10.05 ± 1.35	1.54	0.125	−0.05 (−0.35–0.03)	0.089
Men	339 (32.8)	9.89 ± 1.61			Ref	
Marital status						
Married	512 (49.6)	10.24 ± 1.20	−5.46	<0.001	0.02 (−0.18–0.31)	0.580
Unmarried	521 (50.4)	9.76 ± 1.60			Ref	
Education						
Middle school and below	207 (20.0)	9.57 ± 1.57	13.61	<0.001	−0.13 (−0.79 to –0.12)	<0.05
Associate degree	152 (14.7)	9.92 ± 1.56			−0.05 (–0.56 to −0.13)	0.227
Bachelor degree	390 (37.8)	9.97 ± 1.44			−0.05 (−0.43–0.14)	0.312
Master degree and above	284 (27.5)	10.38 ± 1.15			Ref	
Occupation						
Unemployed	66 (6.4)	9.94 ± 1.31	14.89	<0.001	−0.02 (−0.40–0.29)	0.754
Employed	482 (46.7)	10.06 ± 1.40			–0.04 (−0.42–0.18)	0.430
Student	277 (26.8)	9.58 ± 1.60			−0.11 (−0.71–0.02)	0.063
Teacher	208 (20.1)	10.43 ± 1.16			Ref	
Region of Indonesia						
Eastern Indonesia	14 (1.4)	10.29 ± 1.14	0.42	0.660	0.01 (−0.64–0.85)	0.788
Central Indonesia	62 (6.0)	9.90 ± 1.32			−0.01 (−0.40–0.32)	0.828
Western Indonesia	957 (92.6)	10.00 ± 1.45			Ref	

SD = standard deviation; CI = confidence interval; ^a^ Independent *t* test; ^b^ analysis of variance. Note: Adjusted beta-coefficient (β-Coef) and 95% confidence interval (CI) were estimated using multiple linear regression.

**Table 3 ijerph-18-04473-t003:** Attitude toward COVID-19 by demographic characteristics and knowledge scores (*n* = 1033).

Characteristics	Attitudes, *n* (%)						
	COVID-19 Will Be Successfully Controlled				Indonesia Can Win Against COVID-19		
	Agree	Disagree	Don’t Know	*p* Value	Yes	No	*p* Value
Age ^a^ (years)							
17–29	517 (93.7)	12 (2.2)	23 (4.2)	0.050	529 (95.8)	23 (4.2)	0.527
≥30	457 (95.0)	2 (0.4)	22 (4.6)		457 (95.0)	24 (5.0)	
Sex ^a^							
Women	666 (96.0)	6 (0.9)	22 (3.2)	0.004	671 (96.7)	23 (3.3)	0.006
Men	308 (90.9)	8 (2.4)	23 (6.8)		315 (92.9)	24 (7.1)	
Marital status ^a^							
Married	487 (95.1)	4 (0.8)	21 (4.1)	0.260	489 (95.5)	23 (4.5)	0.930
Unmarried	487 (93.5)	10 (1.9)	24 (4.6)		497 (95.4)	24 (4.6)	
Education ^a^							
Middle school and below	200 (96.6)	2 (1.0)	5 (2.4)	0.707	204 (98.6)	3 (1.4)	0.082
Associate degree	144 (94.7)	1 (0.7)	7 (4.6)		146 (96.1)	6 (3.9)	
Bachelor’s degree	364 (93.3)	7 (1.8)	19 (4.9)		367 (94.1)	23 (5.9)	
Master’s degree and above	266 (93.7)	4 (1.4)	14 (4.9)		269 (94.7)	15 (5.3)	
Occupation ^a^							
Unemployed workers	63 (95.5)	0	3 (4.5)	0.259	64 (97.0)	2 (3.0)	0.098
Employed workers	451 (93.6)	7 (1.5)	24 (5.0)		452 (93.8)	30 (6.2)	
Student	259 (93.5)	7 (2.5)	11 (4.0)		270 (97.5)	7 (2.5)	
Teacher	201 (96.6)	0	7 (3.4)		200 (96.2)	8 (3.8)	
Region of Indonesia ^a^							
Eastern Indonesia	14 (100.0)	0	0	0.713	13 (92.9)	1 (7.1)	0.790
Central Indonesia	60 (96.8)	1 (1.60)	1 (1.60)		60 (96.8)	2 (3.2)	
Western Indonesia	900 (94.0)	13 (1.40)	44 (4.60)		913 (95.4)	44 (4.6)	
Knowledge score ^d^	10.06 (1.29)	10.07 (1.14)	8.60 (3.06)	<0.001 ^b^	10.03 (1.35)	9.28 (2.56)	0.050 ^c^

^a^ chi-square test; ^b^ analysis of variance; ^c^ Independent t test; ^d^ mean (SD).

**Table 4 ijerph-18-04473-t004:** Practice toward COVID-19 based on demographic characteristics and knowledge scores (*n* = 1033).

Characteristics	Practice, *n* (%)					
	Going to a Crowded Place			Wearing a Mask when Leaving Home		
	Yes	No	*p* Value	Yes	No	*p* Value
Age ^a^ (years)						
17–29	126 (22.8)	426 (77.2)	0.571	478 (86.6)	74 (13.4)	0.070
≥30	117 (24.3)	364 (75.7)		434 (90.2)	47 (9.8)	
Sex ^a^						
Women	146 (21.0)	548 (70.9)	0.007	631 (90.9)	63 (9.1)	<0.001
Men	97 (28.6)	242 (71.4)		281 (82.9)	58 (17.1)	
Marital status ^a^						
Married	124 (24.2)	388 (75.8)	0.602	461 (90.0)	51 (10.0)	0.083
Unmarried	119 (22.8)	402 (77.2)		451 (86.6)	70 (13.4)	
Education ^a^						
Middle school and below	35 (16.9)	172 (83.1)	0.030	117 (85.5)	30 (14.5)	0.046
Associate degree	34 (22.4)	118 (77.6)		141 (92.8)	11 (7.2)	
Bachelor’s degree	108 (27.7)	282 (72.3)		336 (86.2)	54 (13.8)	
Master’s degree and above	66 (23.2)	218 (76.8)		258 (90.8)	26 (9.2)	
Occupation ^a^						
Unemployed workers	13 (19.7)	53 (80.3)	0.035	57 (86.4)	9 (13.3)	0.055
Employed workers	133 (27.6)	349 (72.4)		432 (89.6)	50 (10.4)	
Student	53 (19.1)	224 (80.9)		233 (84.1)	44 (15.9)	
Teacher	44 (21.2)	164 (78.8)		190 (91.3)	18 (8.7)	
Region of Indonesia ^a^						
Eastern Indonesia	3 (21.4)	11 (78.6)	0.529	13 (92.9)	1 (7.1)	0.682
Central Indonesia	11 (17.7)	51 (82.3)		53 (85.5)	9 (14.5)	
Western Indonesia	229 (23.9)	728 (76.1)		846 (88.4)	111 (11.6)	
Knowledge score ^c^	9.86 (1.63)	10.04 (1.37)	0.130 ^b^	10.07 (1.32)	9.42 (2.03)	0.001 ^b^

^a^ chi-square; ^b^ Independent t test; ^c^ mean (SD).

**Table 5 ijerph-18-04473-t005:** Hierarchical logistic regression analysis of factors significantly associated with practice toward COVID-19.

Variables	Going to a Crowded Place						Wearing a Mask when Leaving Home					
	Step 1		Step 2		Step 3		Step 1		Step 2		Step 3	
	OR ^b^	*p* Value	OR ^b^	*p* Value	OR ^b^	*p* Value	OR ^b^	*p* Value	OR ^b^	*p* Value	OR ^b^	*p* Value
Age (years)												
17–29	0.99	0.949	0.96	0.843	0.95	0.819	0.91	0.734	0.96	0.892	0.96	0.885
≥30	Ref	-	Ref	-	Ref	-	Ref	-	Ref	-	Ref	-
Sex												
Women	0.71	0.027	0.72	0.037	0.75	0.064	2.11	0.000	2.03	0.001	1.92	0.002
Men	Ref	-	Ref	-	Ref	-	Ref	-	Ref	-	Ref	-
Marital status												
Married	0.95	0.813	0.96	0.833	0.97	0.891	1.01	0.963	0.99	0.958	0.96	0.897
Unmarried	Ref	-	Ref	-	Ref	-	Ref	-	Ref	-	Ref	-
Education												
Middle school and below	0.64	0.123	0.61	0.091	0.66	0.166	0.90	0.783	0.99	0.980	0.89	0.762
Associate degree	0.84	0.548	0.82	0.508	0.87	0.635	1.45	0.408	1.54	0.344	1.46	0.405
Bachelor’s degree	1.10	0.673	1.09	0.708	1.13	0.599	0.81	0.515	0.82	0.557	0.79	0.479
Master’s degree and above	Ref	-	Ref	-	Ref	-	Ref	-	Ref	-	Ref	-
Occupation												
Unemployed workers	1.12	0.772	1.11	0.789	1.07	0.858	0.65	0.388	0.67	0.427	0.70	0.495
Employed workers	1.41	0.163	1.40	0.176	1.34	0.242	1.00	0.997	1.04	0.908	1.10	0.800
Student	1.02	0.958	0.98	0.942	0.95	0.863	0.60	0.211	0.65	0.306	0.66	0.324
Teacher	Ref	-	Ref	-	Ref	-	Ref	-	Ref	-	Ref	-
Region of Indonesia												
Eastern Indonesia	0.68	0.273	0.68	0.269	0.69	0.291	0.85	0.665	0.85	0.671	0.82	0.603
Central Indonesia	0.91	0.884	0.92	0.902	0.92	0.896	1.40	0.748	1.39	0.758	1.42	0.741
Western Indonesia	Ref	-	Ref	-	Ref	-	Ref	-	Ref	-	Ref	-
Knowledge score			0.90	0.046	0.92	0.118			1.26	0.000	1.22	0.002
A1 ^a^: COVID-19 will be successfully controlled												
Agree					0.53	0.275					1.24	0.763
Don’t know					0.69	0.570					0.87	0.862
Disagree	Ref	-	Ref	-	Ref	-	Ref	-	Ref	-	Ref	-
A2 ^a^: Indonesia can win against COVID-19												
Yes					0.60	0.146					2.19	0.062
No	Ref	-	Ref	-	Ref	-	Ref	-	Ref	-	Ref	-

^a^ A = attitude; ^b^ OR = odds ratio.

**Table 6 ijerph-18-04473-t006:** Multivariate logistic regression of factors predicting practice among men and women.

Variables	Going to a Crowded Place				Wearing a Mask When Leaving Home			
	Men (*n* = 339)		Women (*n* = 694)		Men (*n* = 339)		Women (*n* = 694)	
	OR ^b^	*p* Value	OR ^b^	*p* Value	OR ^b^	*p* Value	OR ^b^	*p* Value
Age								
17–29	0.77	0.445	1.08	0.778	0.74	0.490	1.08	0.858
≥30	Ref	-	Ref	-	Ref	-	Ref	-
Marital status								
Married	0.58	0.116	1.44	0.189	0.62	0.269	1.27	0.560
Unmarried	Ref	-	Ref	-	Ref	-	Ref	-
Education								
Middle school and below	0.77	0.598	0.58	0.172	0.49	0.201	1.64	0.399
Associate degree	1.07	0.913	0.77	0.461	1.53	0.602	1.74	0.352
Bachelor’s degree	1.74	0.176	0.85	0.584	0.70	0.481	1.00	0.995
Master’s degree and above	Ref	-	Ref	-	Ref	-	Ref	-
Occupation								
Unemployed workers	0.81	0.824	1.32	0.557	0.54	0.355	0.54	0.355
Employed workers	1.30	0.570	1.36	0.321	1.16	0.769	1.16	0.769
Student	0.86	0.772	1.24	0.607	0.57	0.375	0.57	0.375
Teacher	Ref	-	Ref	-	Ref	-	Ref	-
Region of Indonesia								
Eastern Indonesia	0.62	0.389	0.68	0.390	2.37	0.266	0.49	0.119
Central Indonesia	0.00	0.999	1.41	0.627	0.00	0.999	0.81	0.842
Western Indonesia	Ref	-	Ref	-	Ref	-	Ref	-
Knowledge score	0.79	0.007	1.02	0.833	1.19	0.070	1.25	0.015
A1 ^a^: COVID-19 will be successfully controlled								
Agree	0.46	0.325	0.58	0.559	1.08	0.938	1.43	0.755
Don’t know	0.54	0.484	0.78	0.812	1.11	0.922	0.70	0.781
Disagree	Ref	-	Ref	-	Ref	-	Ref	-
A2 ^a^: Indonesia can win against COVID-19								
Yes	0.67	0.444	0.52	0.195	3.32	0.029	1.60	0.508
No	Ref	-	Ref	-	Ref	-	Ref	-

^a^ A = Attitude; ^b^ OR = Odds ratio.

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
