# Peer review of "Knowledge, Attitude, and Practice of Indonesian Residents toward COVID-19: A Cross-Sectional Survey"

_ijerph, 2021, doi:10.3390/ijerph18094473_

Round 1
Reviewer 1 Report
The authors provide a study about Indonesians' Knowledge attitude and practice toward COVID-19
The proposed study is interesting but there are some points that the authors should better discuss.
The authors should be better described the novelties of their study with respect to existing ones. In particular, the author should discuss limitation and cons of the examined approaches. Furthermore, the authors should provide more details and discussion about the obtained results. The Discussion section also needs to be improved by analyzing the outcome of evaluation section.
I suggest to further analyze more recent approaches about the examined topics. In particular, I suggest the following papers to further investigate graph-based machine learning and multimedia analysis for analyzing knowledge attitude and practice of users:
1) An Epidemiological Neural network exploiting Dynamic Graph Structured Data applied to the COVID-19 outbreak. IEEE Transactions on Big Data.
2) Multimedia story creation on social networks. Future Generation Computer Systems, 86, 412-420.
Finally, I suggest to perform a linguistic revision.
Author Response
We sincerely thanks to the editor and all reviewers for their valuable suggestions and giving us great opportunity to revised our manuscript entitled “Knowledge, attitude, and practice of Indonesian residents toward COVID-19: a cross sectional survey”. We have incorporated all the suggested changes into the manuscript and have highlighted the revised sections. Hereby, our responses and revision based on editor and reviewer’s comments.
"Please see the attachment"

Reviewer 2 Report
The manuscript document that KAP was crucial in assessing public adherence toward COVID-19 in Indonesia. I have the following comments:
- I suggest that authors highlight the incremental contributions by compare their work with this paper “Positive Correlation Between General Public Knowledge and Attitudes Regarding COVID-19 Outbreak 1 Month After First Cases Reported in Indonesia” DOI: https://doi.org/10.1007/s10900-020-00866-0.
- Cite more relevant and recent studies in the introduction and results by following this link: https://scholar.google.com/scholar?hl=en&as_sdt=0%2C5&q=allintitle%3A+covid-19+and++public+knowledge&btnG=.
- Update the COVID-19 pandemic cases and deaths statistics at World level and in Indonesia in the first paragraph of the introduction.
- Authors provide the detail of ethical approval of this study.
- Authors explain why they use different scales for these variables. Authors term variables as scale such as “knowledge scale” instead of “knowledge domain”.
- Results are very confusing:
- Table 1-3 are providing us the frequency distribution and comparison of responses of demographic variables. However, table titles are correlation based such as “Table 1. Correlation between patients’ demographic characteristics and their knowledge score of COVID-19 (n = 1033)”. I did not find the correlation values in these three variables.
- There is no regression analysis for patients’ demographic characteristics and their knowledge score of COVID-19. As table 4 and 5 are providing the analysis of attitudes and practice.
- In the first sentence of the discussion, authors claimed that “No studies have analyzed a population’s KAP toward COVID-19, particularly in Indonesia”. However, they have cited an Indonesian study reference 21. I urge the authors to revise the discussion by including the results of for patients’ demographic characteristics and their knowledge score of COVID-19. Then discuss the KAP results with comparison to reference 21. Also discuss it with other studies in different contexts.
- Conclusion is very short, extend it with revised results and their implications.
Author Response
We sincerely thanks to the editor and all reviewers for their valuable suggestions and giving us great opportunity to revised our manuscript entitled “Knowledge, attitude, and practice of Indonesian residents toward COVID-19: a cross sectional survey”. We have incorporated all the suggested changes into the manuscript and have highlighted the revised sections. Hereby, our responses and revision based on editor and reviewer’s comments.
"Please see the attachment".

Reviewer 3 Report
I had serious concerns about the manuscript "Knowledge, attitude, and practice of Indonesian residents to ward COVID-19: a cross-sectional survey". It has great difficulty in validating the methodological techniques used. Some details call attention to the manuscript and must be remedied before the text can be considered for publication.
- The sample calculation does not make sense and a number of details are missing that may help us to understand how an adult population of 267.7 million inhabitants generated such a small sample size;
- Where were the initial seeds found? which methodological process was used to guarantee generalization? Which epidemiological data collection technique was used?
- Did the authors have 34 duplicate responses? It makes no sense. Indicates that they have not used the appropriate security instruments and that their text has no internal validity and also loses its external validity for the reasons already listed;
- A free translation of a data collection form into another language without a validation or pre-test incurs measurement bias.
Author Response
We sincerely thanks to the editor and all reviewers for their valuable suggestions and giving us great opportunity to revised our manuscript entitled “Knowledge, attitude, and practice of Indonesian residents toward COVID-19: a cross sectional survey”. We have incorporated all the suggested changes into the manuscript and have highlighted the revised sections. Hereby, our responses and revision based on editor and reviewer’s comments.
"Please see the attachment."

Reviewer 4 Report
The study submitted adds to the large body of research on the effects and factors associated with the COVID-19 pandemic. In order to learn about the knowledge, attitude and practice of a representative population of Indonesians about the disease, a validated instrument in Chinese language and hierarchical logistic regression, by which the variables have been introduced in an ordered manner (from greater to lesser theoretical importance), are applied for the analysis of the predictive effect or causality for qualitative variables.
Although the results can effectively contribute to contrast and augment those obtained by similar research over time (Al-Hanawi, 2020; Zhong, 2020), the following improvements are recommended to be addressed:
1. Expert translation in the language of the population to which the instrument was applied is not sufficient to ensure its reliability and validity. It is therefore recommended that quantitative (at least an exploratory factor analysis) and qualitative empirical evidence be provided to ensure its scientific appropriateness in the context of the Indonesian population under study.
2. Further literature review and discussion of the results obtained is recommended. The abundant studies already available in open access on the proposed research problem recommend its deepening and extension.
Author Response

(The authors gave the same response as above.)

Round 2
Reviewer 1 Report
I think that the authors have addressed all my concerns
Author Response
We would like to thank the editor and all reviewers for their valuable suggestions and for providing us the opportunity to revise our manuscript titled “Knowledge, attitude, and practice of Indonesian residents toward COVID-19: A cross sectional survey.” We have incorporated all suggested changes into the manuscript and have written the revised sections in red font. We have provided the following point-by-point responses to the comments of the editor and reviewers.

Reviewer 2 Report
The authors have addressed my comments, but the comments regarding statistical tests and findings of the paper still need significant improvement for publication:
- The paper needs extensive proof editing from an expert.
- Regarding the incremental contribution comment, “Based on the study reference 21 (Sari, 2020), they measured only two variables which are knowledge and attitude, no practice variable included. Therefore, we claimed that no previous studies measured knowledge, attitude and practice (KAP) in Indonesia (Line 226-236, Page 5)”.
I suggest authors do not claim no previous study has done. Instead, authors rewrite it as “our study contributes to the literature or extend the literature by including the practice variable”.
- I urge the authors to revise the methods of the study:
- Descriptive analysis is done by reporting the mean, median, standard deviation, maximum and minimum values of the study variables in section 2.4.
- Authors describe the name of the statistical test from which they have conducted the bivariate analysis, such as the independent t-test or the Fisher exact test in section 2.4.
- I urge the authors to run a Pearson correlation analysis as it is a pre-requisite for the regression analysis. Thus, mention it the section 2.4 statistical analysis.
- In the findings, I urge the authors to include two more tables:
- One table for descriptive statistics includes variable name, mean, median, standard deviation, minimum, and maximum values.
- A Pearson Correlation analysis table is required with the correlation coefficient and p-values of the variables.
Author Response

(The authors gave the same response as above.)

Reviewer 3 Report
The manuscript has serious problems of internal and external validity that occurred in the research and that will not be solved with one or more revisions.
For example, any researcher who works as google forms knows that it is possible to limit the response to an IP or even to email and thus avoid selection bias and duplicate responses. Only this detail is necessary to reject the manuscript, as it indicates that it was restricted to a niche of people, most likely close to the researchers.
The statistical calculation is still strange. What is the assumed prevalence? What population was considered? In general it is not clear. If the small number was necessary, why not invest in ways of generalizing the sample to ensure that it was able to reflect the country?
Not even data is provided on how many participants were obtained from each base.
Thus, I am unable to indicate the manuscript for publication unless the research is redone
Author Response

(The authors gave the same response as above.)

Reviewer 4 Report
Suggestions and recommendations have been addressed. However, further details on the resulting theoretical dimensions (not shown) of the exploratory factor analysis (EFA) are required.
The authors will agree with the reviewr that the demonstration of the possibility of factoring the scale does not yield the result that is sought in terms of validation of the instrument.
Author Response

(The authors gave the same response as above.)
